# Epigenetic scores for the circulating proteome as tools for disease prediction

**Danni A Gadd[1†], Robert F Hillary[1†], Daniel L McCartney[1†], Shaza B Zaghlool[2,3†], Anna J Stevenson[1], Yipeng Cheng[1], Chloe Fawns-Ritchie[1,4], Cliff Nangle[1], Archie Campbell[1], Robin Flaig[1], Sarah E Harris[4,5], Rosie M Walker[6], Liu Shi[7], Elliot M Tucker-Drob[8,9], Christian Gieger[10,11,12,13], Annette Peters[11,12,13], Melanie Waldenberger[10,11,12], Johannes Graumann[14,15], Allan F McRae[16], Ian J Deary[4,5], David J Porteous[1], Caroline Hayward[1,17], Peter M Visscher[16], Simon R Cox[4,5], Kathryn L Evans[1], Andrew M McIntosh[1,18], Karsten Suhre[2], Riccardo E Marioni[1]\***

[1]Centre for Genomic and Experimental Medicine, Institute of Genetics and Cancer, University of Edinburgh, Edinburgh, United Kingdom; [2]Department of Physiology and Biophysics, Weill Cornell Medicine-Qatar, Education City, Doha, Qatar; [3]Computer Engineering Department, Virginia Tech, Blacksburg, United States; [4]Department of Psychology, University of Edinburgh, Edinburgh, United Kingdom; [5]Lothian Birth Cohorts, University of Edinburgh, Edinburgh, United Kingdom; [6]Centre for Clinical Brain Sciences, Chancellor's Building, University of Edinburgh, Edinburgh, United Kingdom; [7]Department of Psychiatry, University of Oxford, Oxford, United Kingdom; [8]Department of Psychology, The University of Texas at Austin, Austin, United States; [9]Population Research Center, The University of Texas at Austin, Austin, United States; [10]Research Unit Molecular Epidemiology, Helmholtz Zentrum München, German Research Center for Environmental Health, Neuherberg, Germany; [11]Institute of Epidemiology, Helmholtz Zentrum München, German Research Center for Environmental Health, Neuherberg, Germany; [12]German Center for Cardiovascular Research (DZHK), partner site Munich Heart Alliance, Munich, Germany; [13]German Center for Diabetes Research (DZD), Neuherberg, Germany; [14]Scientific Service Group Biomolecular Mass Spectrometry, Max Planck Institute for Heart and Lung Research, W.G. Kerckhoff Institute, Bad Nauheim, Germany; [15]German Centre for Cardiovascular Research (DZHK), Partner Site Rhine-Main, Max Planck Institute of Heart and Lung Research, Bad Nauheim, Germany; [16]Institute for Molecular Bioscience, University of Queensland, Brisbane, Australia; [17]Medical Research Council Human Genetics Unit, Institute of Genetics and Cancer, University of Edinburgh, Edinburgh, United Kingdom; [18]Division of Psychiatry, University of Edinburgh, Royal Edinburgh Hospital, Edinburgh, United Kingdom

**\*For correspondence:**
Riccardo.Marioni@ed.ac.uk

†These authors contributed equally to this work

**Abstract** Protein biomarkers have been identified across many age-related morbidities. However, characterising epigenetic influences could further inform disease predictions. Here, we leverage epigenome-wide data to study links between the DNA methylation (DNAm) signatures of the circulating proteome and incident diseases. Using data from four cohorts, we trained and tested epigenetic scores (EpiScores) for 953 plasma proteins, identifying 109 scores that explained between 1% and 58% of the variance in protein levels after adjusting for known protein quantitative trait loci (pQTL) genetic effects. By projecting these EpiScores into an independent sample (Generation Scotland; n = 9537) and relating them to incident morbidities over a follow-up of 14 years, we uncovered

130 EpiScore-disease associations. These associations were largely independent of immune cell proportions, common lifestyle and health factors, and biological aging. Notably, we found that our diabetes-associated EpiScores highlighted previous top biomarker associations from proteome-wide assessments of diabetes. These EpiScores for protein levels can therefore be a valuable resource for disease prediction and risk stratification.

## Editor's evaluation
This is an important study that demonstrates the potential utility of the circulating proteome for disease prediction and risk stratification.

## Introduction
Chronic morbidities place longstanding burdens on our health as we age. Stratifying an individual's risk prior to symptom presentation is therefore critical (*NHS England, 2016*). Though complex morbidities are partially driven by genetic factors (*Fuchsberger et al., 2016*; *Yao et al., 2018*), epigenetic modifications have also been associated with disease (*Lord and Cruchaga, 2014*). DNA methylation (DNAm) encodes information on the epigenetic landscape of an individual and blood-based DNAm signatures have been found to predict all-cause mortality and disease onset, providing strong evidence to suggest that methylation is an important measure of disease risk (*Hillary et al., 2020a*; *Lu et al., 2019*; *Zhang et al., 2017*). DNAm can regulate gene transcription (*Lea et al., 2018*), and epigenetic differences can be reflected in the variability of the proteome (*Hillary et al., 2019*; *Hillary et al., 2020b*; *Zaghlool et al., 2020*). Low-grade inflammation, which is thought to exacerbate many age-related morbidities, is particularly well captured through DNAm studies of plasma protein levels (*Zaghlool et al., 2020*). As proteins are the primary effectors of disease, connecting the epigenome, proteome, and time to disease onset may help to resolve predictive biological signatures.

Epigenetic predictors have utilised DNAm from the blood to estimate a person's 'biological age' (*Lu et al., 2019*), measure their exposure to lifestyle and environmental factors (*McCartney et al., 2018c*; *McCartney et al., 2018a*; *Peters et al., 2021*), and predict circulating levels of inflammatory proteins (*Stevenson et al., 2020*; *Stevenson et al., 2021*). A leading epigenetic predictor of biological aging, the GrimAge epigenetic clock incorporates methylation scores for seven proteins along with smoking and chronological age, and is associated with numerous incident disease outcomes independently of smoking (*Hillary et al., 2020a*; *Lu et al., 2019*). This suggests there is predictive value gained in utilising DNAm scores relevant to protein levels as intermediaries for predictions. Methylation scores also point towards the pathways that may act on health beyond the protein biomarker that they are trained on. A portfolio of methylation scores for proteins across the circulating proteome could therefore aid in the prediction of disease and offer a different, but additive signal beyond methylation or protein data alone. Generation of an extensive range of epigenetic scores for protein levels has not been attempted to date. The capability of specific protein scores to predict a range of morbidities has also not been tested. However, DNAm scores for interleukin-6 (IL-6) and C-reactive protein (CRP) have been found to associate with a range of phenotypes independently of measured protein levels, show more stable longitudinal trajectories than repeated protein measurements, and, in some cases, outperform blood-based proteomic associations with brain morphology (*Stevenson et al., 2021*; *Conole et al., 2021*). This is likely due to DNAm representing the accumulation of more sustained effects over a longer period of time than protein measurements, which have often been shown to be variable in their levels when measured at multiple time points (*Koenig et al., 2003*; *Liu et al., 2015*; *Moldoveanu et al., 2000*; *Shah et al., 2014*). DNAm scores for proteins could therefore be used to alert clinicians to individuals with high-risk biological signatures, many years prior to disease onset.

Here, we report a comprehensive association study of blood-based DNAm with proteomics and disease (*Figure 1*). We trained epigenetic scores – referred to as EpiScores – for 953 plasma proteins (with sample size ranging from 706 to 944 individuals) and validated them using two independent cohorts with 778 and 162 participants. We regressed out known genetic pQTL effects from the protein levels prior to generating the EpiScores to preclude the signatures being driven by common SNP data that are invariant across the lifespan. We then examined whether the most robust predictors (n = 109 EpiScores) associated with the incidence of 12 major morbidities (*Table 1*), over a follow-up period of

**eLife digest** Although our genetic code does not change throughout our lives, our genes can be turned on and off as a result of epigenetics. Epigenetics can track how the environment and even certain behaviors add or remove small chemical markers to the DNA that makes up the genome. The type and location of these markers may affect whether genes are active or silent, this is, whether the protein coded for by that gene is being produced or not. One common epigenetic marker is known as DNA methylation. DNA methylation has been linked to the levels of a range of proteins in our cells and the risk people have of developing chronic diseases.

Blood samples can be used to determine the epigenetic markers a person has on their genome and to study the abundance of many proteins. Gadd, Hillary, McCartney, Zaghlool et al. studied the relationships between DNA methylation and the abundance of 953 different proteins in blood samples from individuals in the German KORA cohort and the Scottish Lothian Birth Cohort 1936. They then used machine learning to analyze the relationship between epigenetic markers found in people's blood and the abundance of proteins, obtaining epigenetic scores or 'EpiScores' for each protein. They found 109 proteins for which DNA methylation patterns explained between at least 1% and up to 58% of the variation in protein levels.

Integrating the 'EpiScores' with 14 years of medical records for more than 9000 individuals from the Generation Scotland study revealed 130 connections between EpiScores for proteins and a future diagnosis of common adverse health outcomes. These included diabetes, stroke, depression, various cancers, and inflammatory conditions such as rheumatoid arthritis and inflammatory bowel disease.

Age-related chronic diseases are a growing issue worldwide and place pressure on healthcare systems. They also severely reduce quality of life for individuals over many years. This work shows how epigenetic scores based on protein levels in the blood could predict a person's risk of several of these diseases. In the case of type 2 diabetes, the EpiScore results replicated previous research linking protein levels in the blood to future diagnosis of diabetes. Protein EpiScores could therefore allow researchers to identify people with the highest risk of disease, making it possible to intervene early and prevent these people from developing chronic conditions as they age.

up to 14 years in the Generation Scotland cohort (n = 9537). We also tested for associations between EpiScore levels and COVID-19 disease outcomes. We regressed out the effects of age on protein levels prior to training and testing; age was also included as a covariate in disease prediction models. We controlled for common risk factors for disease and assessed the capacity of EpiScores to identify previously reported protein-disease associations.

Our MethylDetectR shiny app (*Hillary and Marioni, 2020*) has CpG weights for the 109 EpiScores integrated such that it automates the process of score generation for any DNAm dataset and is available at: https://www.ed.ac.uk/centre-genomic-medicine/research-groups/marioni-group/methyl-detectr. A video on how to use the MethylDetectR shiny app to generate EpiScores is available at: https://youtu.be/65Y2Rv-4tPU.

## Results

### Selecting the most robust EpiScores for protein levels

To generate epigenetic scores for a comprehensive set of plasma proteins, we ran elastic net penalised regression models using protein measurements from the SOMAscan (aptamer-based) and Olink (antibody-based) platforms. We used two cohorts: the German population-based study KORA (n = 944, mean age 59 years [SD 7.8], with 793 SOMAscan proteins) and the Scottish Lothian Birth Cohort 1936 (LBC1936) study (between 706 and 875 individuals in the training cohort, with a total of 160 Olink neurology and inflammatory panel proteins). The mean age of the LBC1936 participants at sampling was 70 (SD 0.8) for inflammatory and 73 (SD 0.7) for neurology proteins. Full demographic information is available for all cohorts in *Supplementary file 1A*.

Prior to running the elastic net models, we rank-based inverse normalised protein levels and adjusted for age, sex, cohort-specific variables and, where present, *cis* and *trans* pQTL effects identified from previous analyses (*Hillary et al., 2019*; *Hillary et al., 2020b*; *Suhre et al., 2017*)

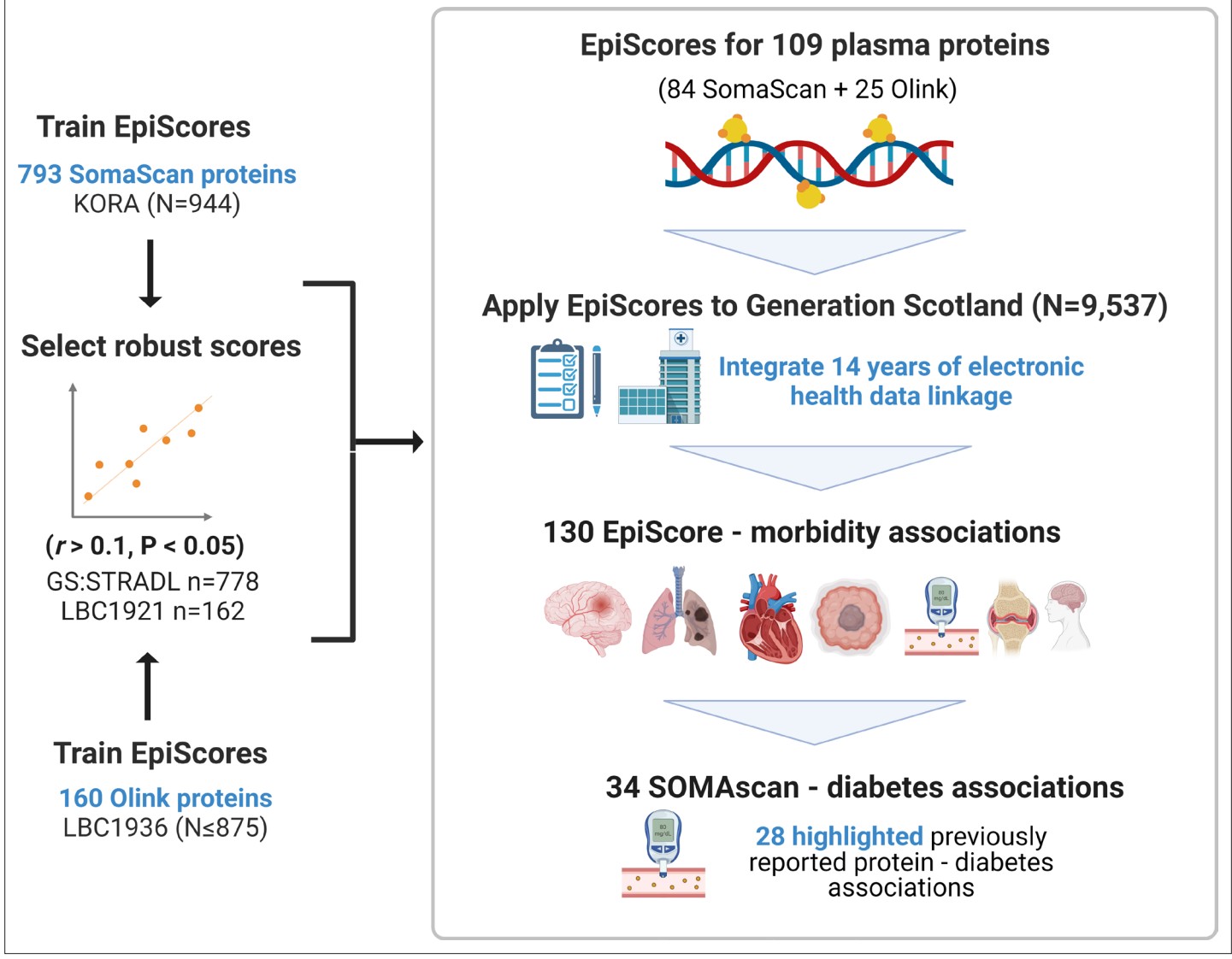

**Figure 1.** EpiScores for plasma proteins as tools for disease prediction study design. DNA methylation scores were trained on 953 circulating plasma protein levels in the KORA and LBC1936 cohorts. There were 109 EpiScores selected based on performance ($r > 0.1$, $p < 0.05$) in independent test sets. The selected EpiScores were projected into Generation Scotland, a cohort that has extensive data linkage to GP and hospital records. We tested whether levels of each EpiScore at baseline could predict the onset of 12 leading causes of morbidity, over a follow-up period of up to 14 years; 130 EpiScore-disease associations were identified, for 10 morbidities. We then assessed whether EpiScore associations reflected protein associations for diabetes, which is a trait that has been well characterised using SOMAscan protein measurements. Of the 34 SOMAscan-derived EpiScore-diabetes associations, 28 highlighted previously reported protein-diabetes associations.

(Materials and methods). Of a possible 793 proteins in KORA, 84 EpiScores had Pearson $r > 0.1$ and $p < 0.05$ when tested in an independent subset of Generation Scotland (The Stratifying Resilience and Depression Longitudinally [STRADL] study, n = 778) (*Supplementary file 1B*). These EpiScores were selected for EpiScore-disease analyses. Of the 160 Olink proteins trained in LBC1936, there were 21 with $r > 0.1$ and $p < 0.05$ in independent test sets (STRADL, n = 778, Lothian Birth Cohort 1921: LBC1921, n = 162) (*Supplementary file 1C*). Independent test set data were not available for four Olink proteins. However, they were included based on their performance ($r > 0.1$ and $p < 0.05$) in a holdout sample of 150 individuals who were left out of the training set. We then retrained all selected predictors on the full training samples.

A total of 109 EpiScores (84 SOMAscan-based and 25 Olink-based) were brought forward ($r > 0.1$ and $p < 0.05$) to EpiScore-disease analyses (*Figure 2* and *Supplementary file 1D*). There were five EpiScores for proteins common to both Olink and SOMAscan panels, which had variable correlation

**Table 1. Incident morbidities in the Generation Scotland cohort**.
Counts are provided for the number of cases and controls for each incident trait in the basic and fully adjusted Cox models run in the Generation Scotland cohort (n = 9537). Mean time-to-event is summarised in years for each phenotype. Alzheimer's dementia cases and controls were restricted to those older than 65 years. Breast cancer cases and controls were restricted to females.

| Morbidity | Basic model | | | Fully adjusted model | | |
|---|---|---|---|---|---|---|
| | N cases | N controls | Years to event (mean, SD) | N cases | N controls | Years to event (mean, SD) |
| Rheumatoid arthritis | 63 | 9289 | 5.6 (3.5) | 52 | 7742 | 6.1 (3.3) |
| Alzheimer's dementia | 69 | 3764 | 7.7 (3) | 52 | 3137 | 7.6 (3.1) |
| Bowel cancer | 78 | 9398 | 6.4 (3.2) | 66 | 7817 | 6.5 (3.2) |
| Depression | 95 | 8317 | 4 (3.2) | 75 | 6984 | 3.8 (3.2) |
| Lung cancer | 100 | 9433 | 5.6 (3.2) | 78 | 7850 | 5.6 (3.1) |
| Breast cancer | 131 | 5356 | 6.1 (3.4) | 111 | 4402 | 5.9 (3.4) |
| Inflammatory bowel disease | 194 | 9114 | 5 (3.6) | 155 | 7592 | 4.8 (3.6) |
| Stroke | 313 | 9026 | 6.4 (3.4) | 246 | 7547 | 6.3 (3.5) |
| COPD | 322 | 8960 | 5.5 (3.4) | 253 | 7476 | 5.5 (3.5) |
| Ischaemic heart disease | 385 | 8649 | 5.6 (3.4) | 302 | 7251 | 5.7 (3.4) |
| Diabetes | 429 | 8757 | 5.6 (3.4) | 322 | 7332 | 5.5 (3.4) |
| Pain | 1329 | 5480 | 4.8 (3.5) | 1081 | 4593 | 4.9 (3.5) |

COPD: chronic obstructive pulmonary disease.

strength (GZMA $r = 0.71$, MMP.1 $r = 0.46$, CXCL10 $r = 0.35$, NTRK3 $r = 0.26$, and CXCL11 $r = 0.09$). Predictor weights, positional information, and *cis/trans* status for CpG sites contributing to these EpiScores are available in *Supplementary file 1E*. The number of CpG features selected for EpiScores ranged from 1 (lyzozyme) to 395 (aminoacylase-1 [ACY-1]), with a mean of 96 (*Supplementary file 1F*). The most frequently selected CpG was the smoking-related site cg05575921 (mapping to the *AHRR* gene), which was included in 25 EpiScores. Counts for each CpG site are summarised in *Supplementary file 1G*. This table includes the set of protein EpiScores that each CpG contributes to, along with phenotypic annotations (traits) from the MRC-IEU EWAS catalog (*MRC-IEU, 2021*) for each CpG site having genome-wide significance ($p < 3.6 \times 10^{-8}$) (*Saffari et al., 2018*). GeneSet enrichment analysis of the original proteins used to train the 109 EpiScores highlighted pathways associated with immune response and cell remodelling, adhesion, and extracellular matrix function (*Supplementary file 1H*).

## EpiScore-disease associations in Generation Scotland

The Generation Scotland dataset contains extensive electronic health data from GP and hospital records as well as DNAm data for 9537 individuals. This makes it uniquely positioned to test whether EpiScore signals can predict disease onset. We ran nested mixed effects Cox proportional hazards models (*Figure 3*) to determine whether the levels of each EpiScore at baseline associated with the incidence of 12 morbidities over a maximum of 14 years of follow-up. The correlation structures for the 109 EpiScore measures used for Cox modelling are presented in *Figure 2—figure supplement 1*.

There were 286 EpiScore-disease associations with a false discovery rate (FDR)-adjusted p < 0.05 in the basic model. After further adjustment for common risk factor covariates (smoking, social deprivation status, educational attainment, body mass index [BMI], and alcohol consumption), 130 of the 286 EpiScore-disease associations from the basic model had p < 0.05 in the fully adjusted model (*Supplementary file 1I-J*). Ten of the 130 fully adjusted associations failed the Cox proportional hazards assumption for the EpiScore variable (p < 0.05 for the association between the Schoenfeld residuals and time; *Supplementary file 1K*). When we restricted the time-to-event/censor period by each year

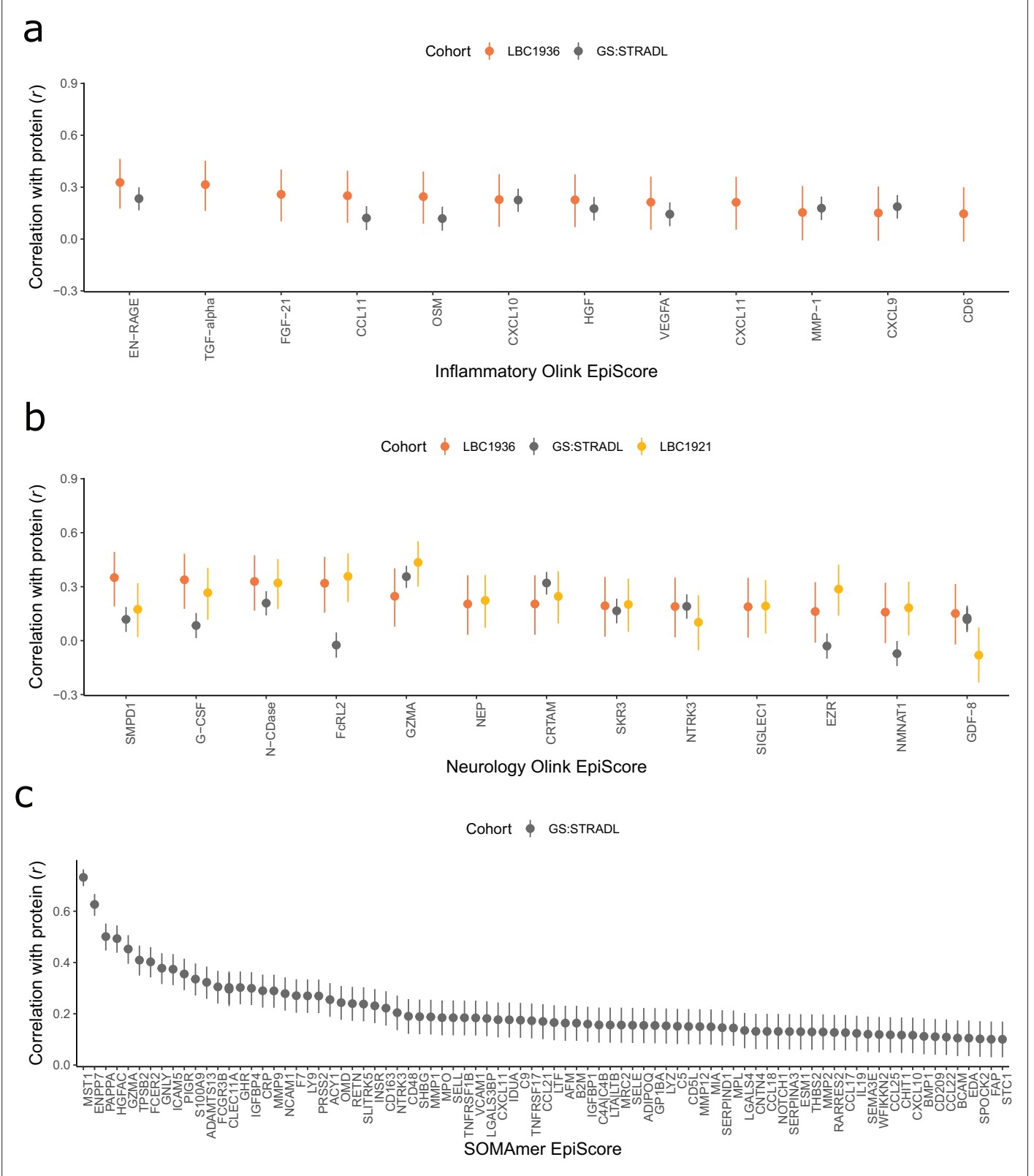

**Figure 2.** Test performance for the 109 selected protein EpiScores. Test set correlation coefficients for associations between protein EpiScores for (**a**) inflammatory Olink, (**b**) neurology Olink, and (**c**) SOMAmer protein panel EpiScores and measured protein levels are plotted. 95% confidence intervals are shown for each correlation. The 109 protein EpiScores shown had $r > 0.1$ and $p < 0.05$ in either one or both of the GS:STRADL (n = 778) and LBC1921 (n = 162) test sets, wherever protein data was available for comparison. Data shown corresponds to the results included in *Supplementary file*

*Figure 2 continued on next page*

*Figure 2 continued*

*1B-C*. Correlation heatmaps between the 109 EpiScore measures (*Figure 2—figure supplement 1*) are provided, along with a summary of the most enriched functional pathways for the genes of the 109 proteins used to train EpiScores (*Figure 2—figure supplement 2*).

The online version of this article includes the following figure supplement(s) for figure 2:

**Figure supplement 1.** Correlation heatmap for protein EpiScore measures in Generation Scotland.

**Figure supplement 2.** GeneSet enrichment of canonical pathways common to the genes encoding proteins that were used to train the 109 selected EpiScores.

of possible follow-up, there were minimal differences in the EpiScore-disease hazard ratios between follow-up periods that did not violate the assumption and those that did (*Supplementary file 1L*). The 130 associations were therefore retained as the primary results.

The 130 associations found in the fully adjusted model comprised 70 unique EpiScores that were related to the incidence of 10 of the 12 morbidities studied. Diabetes and chronic obstructive pulmonary disease (COPD) had the greatest number of associations, with 38 and 37, respectively. *Figure 4* presents the EpiScore-disease relationships for COPD and the remaining nine morbidities: stroke, lung cancer, ischaemic heart disease (IHD), inflammatory bowel disease (IBD), rheumatoid arthritis (RA), depression, bowel cancer and pain (back/neck). There were 16 EpiScores that associated with the onset of three or more morbidities. *Figure 5* presents relationships for these 16 EpiScores in the fully adjusted Cox model results. Of note is the EpiScore for Complement 5 (C5), which associated with four outcomes: stroke, diabetes, RA and COPD. Of the 34 SOMAscan-derived EpiScore associations with incident diabetes, 28 replicated previously reported SOMAscan protein associations (*Elhadad et al., 2020*; *Gudmundsdottir et al., 2020*; *Ngo et al., 2021*) with incident or prevalent diabetes in one or more cohorts (*Figure 6* and *Supplementary file 1M*).

## Immune cell and GrimAge sensitivity analyses

Correlations of the 70 EpiScores that were associated with incident disease ($P < 0.05$ in the fully-adjusted cox proportional hazards models) with covariates suggested interlinked relationships with both estimated white blood cell proportions and GrimAge acceleration (*Figure 3—figure supplement 1*). These covariates were therefore added incrementally to the fully-adjusted Cox models (*Figure 3*). There were 99 associations that remained statistically significant (FDR $p < 0.05$ in the basic model

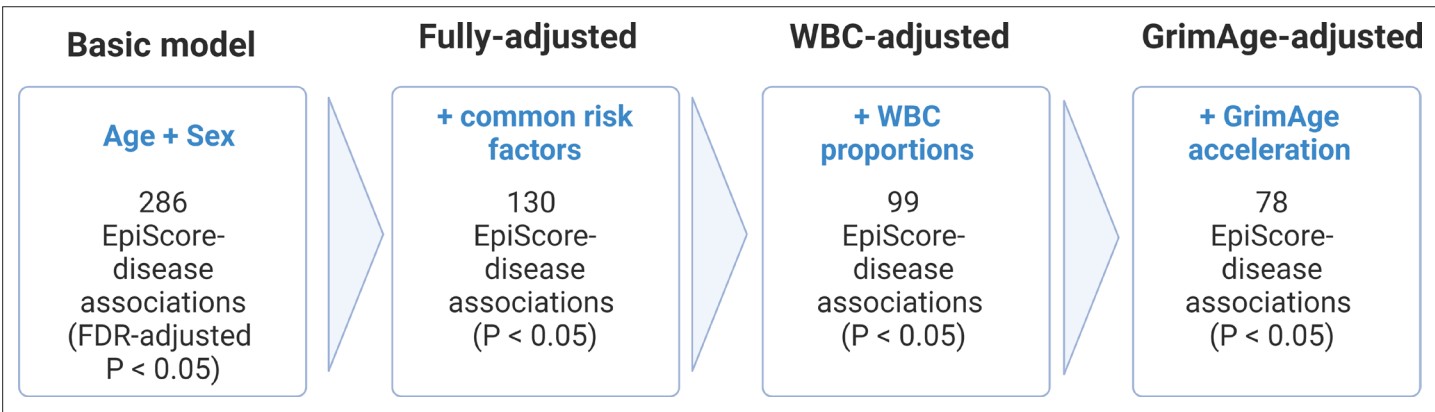

**Figure 3.** Nested Cox proportional hazards assessment of protein EpiScore-disease prediction. Mixed effects Cox proportional hazards analyses in Generation Scotland (n = 9537) tested the relationships between each of the 109 selected EpiScores and the incidence of 12 leading causes of morbidity (*Supplementary file 1I-J*). The basic model was adjusted for age and sex and yielded 286 associations between EpiScores and disease diagnoses, with false discovery rate (FDR)-adjusted $p < 0.05$. In the fully adjusted model, which included common risk factors as additional covariates (smoking, deprivation, educational attainment, body mass index (BMI), and alcohol consumption), 130 of the basic model associations remained significant with $p < 0.05$. In a sensitivity analysis, the addition of estimated white blood cells (WBCs) to the fully adjusted models led to the attenuation of 31 of the 130 associations. In a further sensitivity analysis, 78 associations remained after adjustment for both immune cell proportions and GrimAge acceleration.

The online version of this article includes the following figure supplement(s) for figure 3:

**Figure supplement 1.** Phenotypic trait and estimated white blood cell proportion correlations with EpiScores.

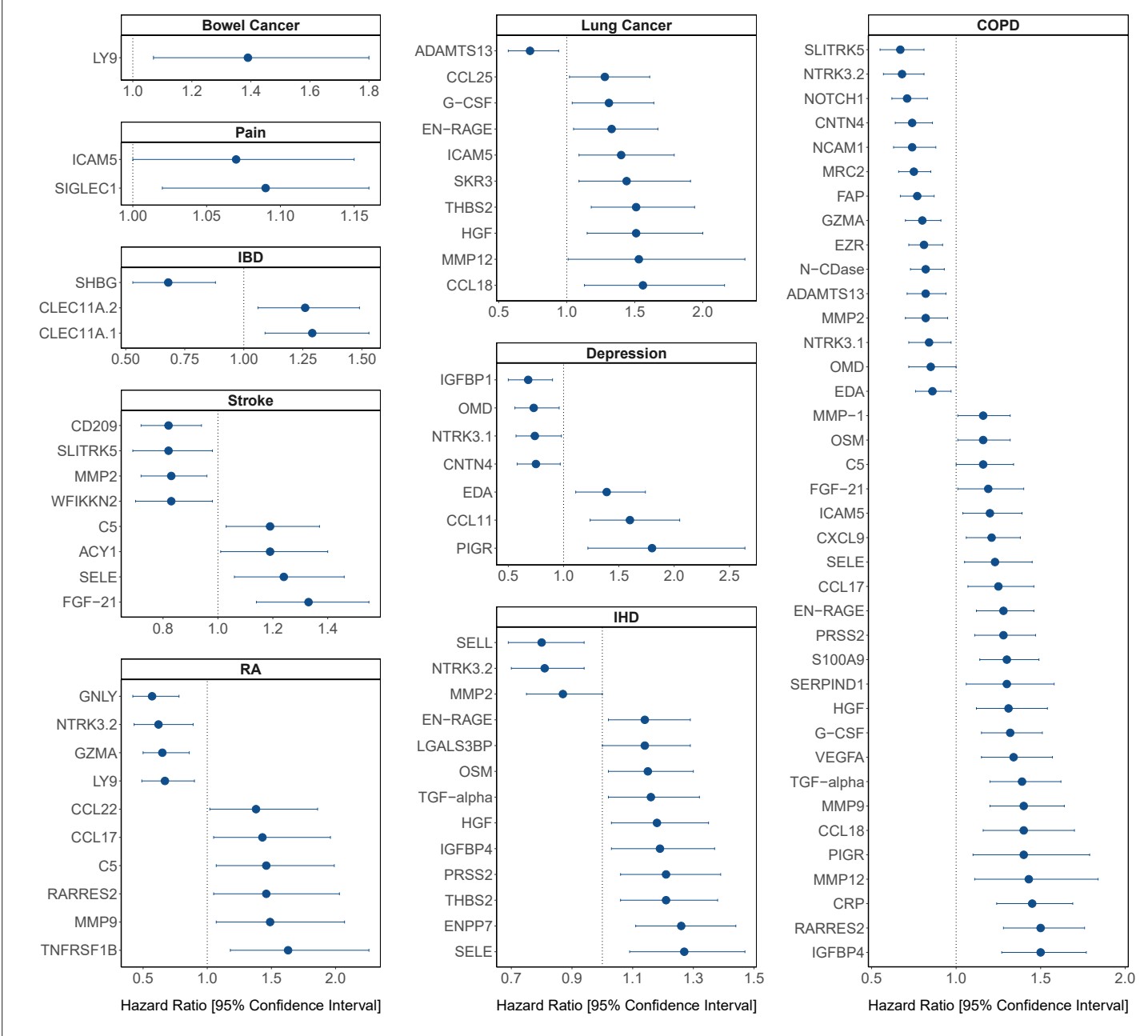

**Figure 4.** Protein EpiScore associations with incident disease. EpiScore-disease associations for 9 of the 11 morbidities with associations where p < 0.05 in the fully adjusted mixed effects Cox proportional hazards models in Generation Scotland (n = 9537). Hazard ratios are presented with confidence intervals for 92 of the 130 EpiScore-incident disease associations reported. Models were adjusted for age, sex, and common risk factors (smoking, body mass index (BMI), alcohol consumption, deprivation, and educational attainment). IBD: inflammatory bowel disease. IHD: ischaemic heart disease. COPD: chronic obstructive pulmonary disease. For EpiScore-diabetes associations, see *Figure 6*. Data shown corresponds to the results included in *Supplementary file 1J*.

and p < 0.05 in the fully adjusted model) after adjustment for immune cell proportions, of which 78 remained significant when GrimAge acceleration scores were added to this model (*Supplementary file 1J*). In a further sensitivity analysis, relationships between both estimated white blood cell (WBC) proportions and GrimAge acceleration scores with incident diseases were assessed in the Cox model structure independently of EpiScores. Of the 60 possible relationships between WBC measures and the morbidities assessed, three were statistically significant (FDR-adjusted p < 0.05) in the basic model and remained significant with p < 0.05 in the fully adjusted model (*Supplementary file 1N*). A higher

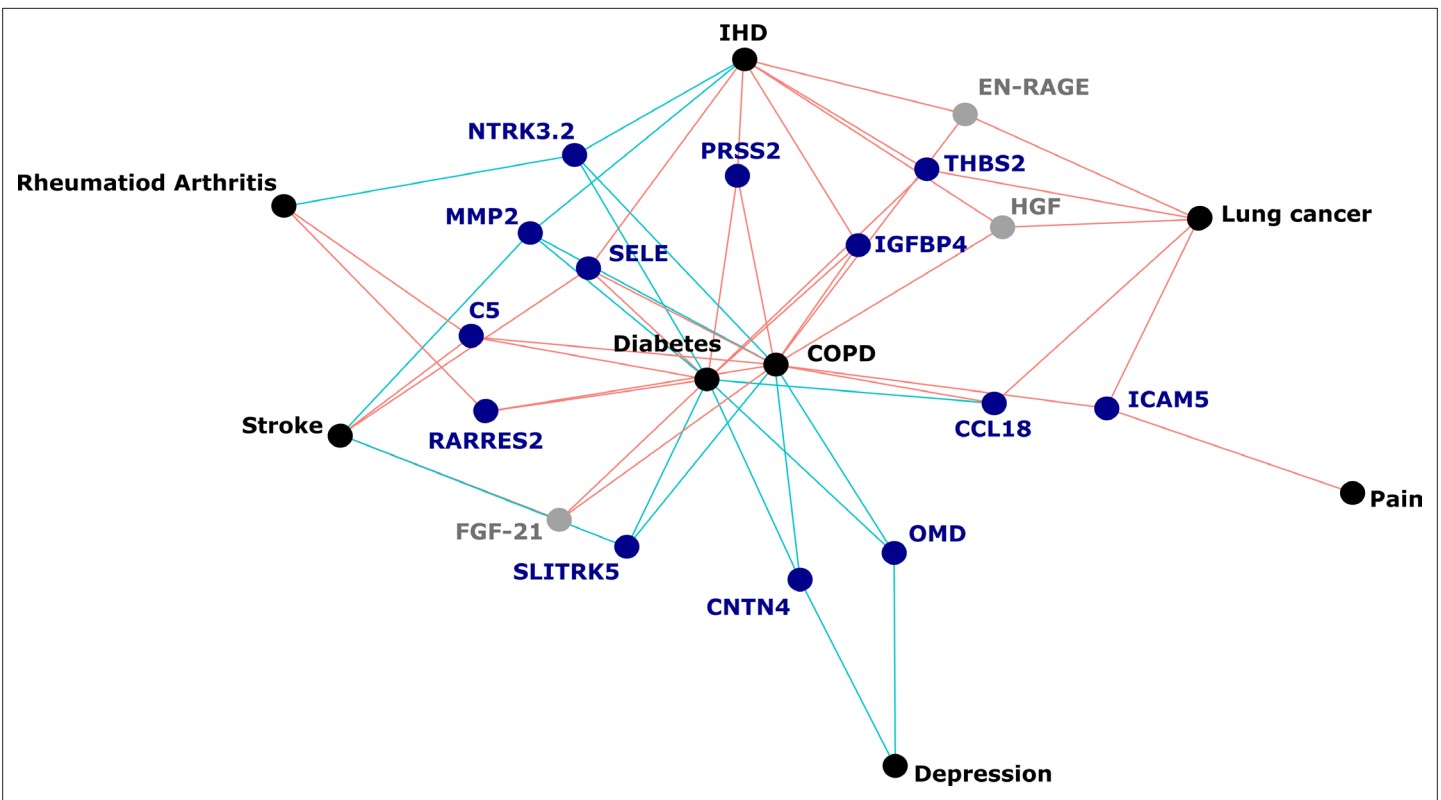

**Figure 5.** Protein EpiScores that associated with the greatest number of morbidities. EpiScores with a minimum of three relationships with incident morbidities in the fully adjusted Cox models. The network includes 16 EpiScores as dark blue (SOMAscan) and grey (Olink) nodes, with disease outcomes in black. EpiScore-disease associations with hazard ratios < 1 are shown as blue connections, whereas hazard ratios > 1 are shown in red. COPD: chronic obstructive pulmonary disease. IHD: ischaemic heart disease. Data shown corresponds to the results included in *Supplementary file 1J*.

proportion of natural killer cells was linked to decreased risk of incident COPD, RA and diabetes. The GrimAge acceleration composite score was associated with COPD, lung cancer, IBD, diabetes and RA in the fully adjusted models (p < 0.05) (*Supplementary file 1O*). The magnitude of the GrimAge effect sizes was comparable to the EpiScore findings.

### Relationship between EpiScores and subsequent COVID-19

Two previous studies including pilot proteomic measurements from the Generation Scotland cohort (N = 199 controls) as part of wider analyses found that several proteins corresponding to our EpiScores were associated with COVID-19 outcomes (*Demichev et al., 2021*; *Messner et al., 2020*). These included proteins such as CRP, C9, SELL, and SHBG, all of which were associated with one or more incident diseases in this study. Two subsets (N = 268 and N = 173) of the Generation Scotland sample who contracted COVID-19 were therefore used to test the hypothesis that EpiScores would associate with COVID-19 outcomes (acquired >9 years after the blood draw for DNAm analyses). No significant associations were identified that delineated differences between the development of long-covid (duration >4 weeks) or hospitalisation from COVID-19 (associations that had p < 0.05 did not withstand Bonferroni adjustment for multiple testing) (*Supplementary file 1P*).

### Discussion

Here, we report a comprehensive DNAm scoring study of 953 circulating proteins. We define 109 robust EpiScores for plasma protein levels that are independent of known pQTL effects. By projecting these EpiScores into a large cohort with extant data linkage, we show that 70 EpiScores associate with the incidence of 10 leading causes of morbidity (130 EpiScore-disease associations in total), but do not associate with COVID-19 outcomes. Finally, we show that EpiScore-diabetes associations highlight previously measured protein-diabetes relationships. The bulk of EpiScore-disease associations are independent of common

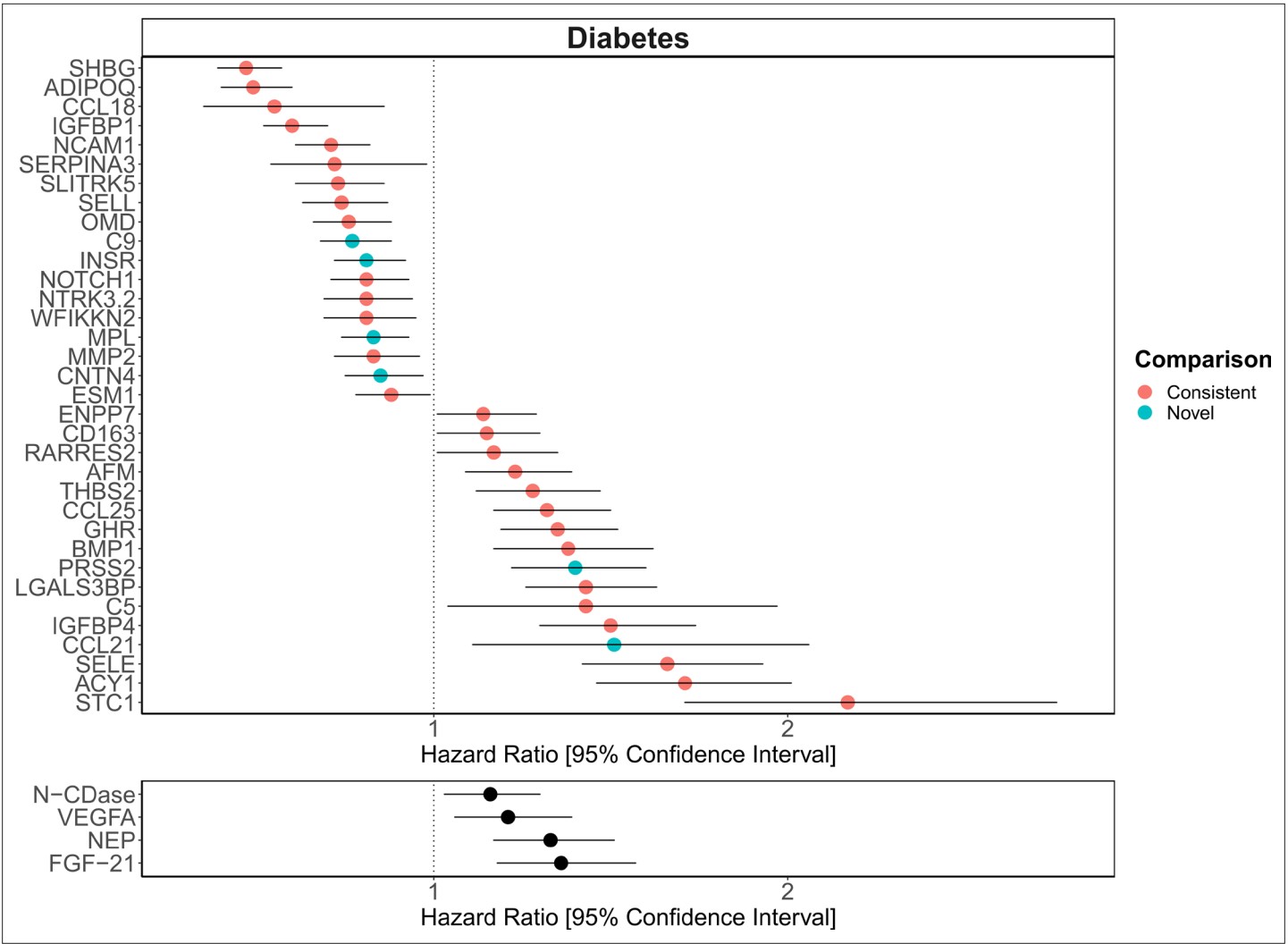

**Figure 6.** Replication of known protein-diabetes associations with protein EpiScores. EpiScore-incident diabetes associations in Generation Scotland (n = 9537). The 34 SOMAscan (top panel) and four Olink (bottom panel) associations shown with p < 0.05 in fully adjusted mixed effects Cox proportional hazards models. Of the 34 SOMAscan-derived EpiScores, 28 associations were consistent with protein-diabetes associations (pink) in one or more of the comparison studies that used SOMAscan protein levels. Six associations were novel (blue). Data shown corresponds to the results included in *Supplementary files 1J and M*.

lifestyle and health factors, differences in immune cell composition and GrimAge acceleration. EpiScores therefore provide methylation-proteomic signatures for disease prediction and risk stratification.

The consistency between our EpiScore-diabetes associations and previously identified protein-diabetes relationships (*Elhadad et al., 2020*; *Gudmundsdottir et al., 2020*; *Ngo et al., 2021*) suggests that epigenetic scores identify disease-relevant biological signals. In addition to the comprehensive lookup of SOMAscan proteins with diabetes, several of the markers we identified for COPD and IHD also reflect previous associations with measured proteins (*Ganz et al., 2016*; *Serban et al., 2021*). The three studies used for the diabetes comparison represent the largest candidate protein characterisations of type 2 diabetes to date and the top markers identified included aminoacylase-1 (ACY-1), sex hormone-binding globulin (SHBG) and growth hormone receptor (GHR) (*Elhadad et al., 2020*; *Gudmundsdottir et al., 2020*; *Ngo et al., 2021*). Our EpiScores for these top markers were also associated with diabetes, in addition to EpiScores for several other protein markers reported in these studies. A growing body of evidence suggests that type 2 diabetes is mediated by genetic and epigenetic regulators (*Kwak and Park, 2016*) and proteins such as ACY-1 and GHR are thought to influence a range of diabetes-associated metabolic mechanisms (*Kim and Park, 2017*; *Pérez-Pérez et al., 2012*). Proteins that we identify through EpiScore associations, such as NTR domain-containing protein 2 (WFIKKN2), have also been causally implicated in type 2 diabetes onset

through Mendelian randomisation analysis (*Ngo et al., 2021*). In the case of diabetes, EpiScores may therefore be used as disease-relevant risk biomarkers, many years prior to onset. Validation should be tested when sufficient data become available for the remaining morbidities.

With modest test set performances (e.g., SHBG $r = 0.18$ and ACY-1 $r = 0.25$), it is perhaps surprising that such strong synergy is observed between EpiScores for proteins that associated with diabetes and the trends seen with measured proteins. Nonetheless, DNAm scores for CRP and IL-6 have previously been shown to perform modestly in test sets ($r \sim 0.2$, equivalent to ~4% explained variance in protein level), but augment and often outperform the measured protein related to a range of phenotypes (*Stevenson et al., 2020*; *Stevenson et al., 2021*). Compared to scores utilising DNAm for the prediction of singular diseases, our EpiScores enable the granular study of individual protein predictor signatures with clinical outcomes.

Our large-scale assessment of EpiScores provides a platform for future studies, as composite predictors for traits may be created using our EpiScore database. These should be tested in incident disease predictions when sufficient case data are available. Our results indicated that the set of 109 EpiScores are likely to be heavily enriched for inflammatory, complement system and innate immune system pathways, in addition to extracellular matrix, cell remodelling, and cell adhesion pathways. This reinforces previous work linking chronic inflammation and the epigenome (*Zaghlool et al., 2020*). It also suggests that EpiScores could be useful in the prediction of morbidities that are characterised by differential inflammatory states. An example of this is the EpiScore for Complement Component 5 (C5), which was associated with the onset of four morbidities (*Figure 5*). The EpiScore for C5 is likely to reflect the biological pathways occurring in individuals with heightened complement cascade activity and could be utilised to alert clinicians to individuals at high risk of multimorbidity. Elevated levels of C5 peptides have been associated with severe inflammatory, autoimmune, and neurodegenerative states (*Ma et al., 2019*; *Mantovani et al., 2014*; *Morgan and Harris, 2015*) and a range of C5-targetting therapeutic approaches are in development (*Alawieh et al., 2018*; *Brandolini et al., 2019*; *Hawksworth et al., 2017*; *Hernandez et al., 2017*; *Morgan and Harris, 2015*; *Ort et al., 2020*).

Though EpiScores such as C5 – which occupy central hubs in the disease prediction framework – may provide evidence of early methylation signatures common to the onset of multiple diseases, we did not observe associations between EpiScores and COVID-19 hospitalisation or long-COVID status. This is perhaps surprising, given that many of the morbidities that our EpiScores predicted are also known risk factors for increased risk of death due to COVID-19 (*Williamson et al., 2020*). Many of the proteins corresponding to EpiScores in our study were also associated with COVID-19 severity and progression in two previous studies that included a pilot sample (N = 199) from the Generation Scotland cohort at baseline as control data (*Demichev et al., 2021*; *Messner et al., 2020*). COVID-19 likely has multiple intersecting risk factors that impact severity and recovery, and the lack of associations we observe is likely to be in part due to the limited number of COVID-19 cases available in Generation Scotland. Additionally, there is a large lag time between baseline biological measurement and COVID-19 in our analyses, whereas the two studies that found protein marker associations integrated protein measurements longitudinally and from samples extracted during COVID-19 progression. With increased power available through continued data linkage, EpiScore relationships with COVID-19 outcomes may be observed in future work.

This study has several limitations. First, we demonstrate that EpiScores carry disease-relevant signals that may be clinically meaningful to delineate early disease risk when comparing relative differences within a cohort. However, projecting a new individual onto a reference set is complicated due to absolute differences in methylation quantification resulting from batch and processing effects. Second, future studies should assess paired protein and EpiScore contributions to traits, as inference from EpiScores alone, while useful for disease risk stratification, is not sufficient to determine mechanisms. This may also highlight EpiScores that outperform the measured protein equivalent in disease. Third, the epitope nature of the protein measurement in the SOMAscan panel may incur probe cross-reactivity and non-specific binding; there may also be differences in how certain proteins are measured across panels (*Pietzner et al., 2020*; *Sun et al., 2018*). Comparisons of multiple protein measurement technologies on the same samples would help to explore this in more detail. Fourth, there may be pQTLs with small effect sizes that were not regressed from the proteins prior to generating the EpiScores. Fifth, while training and testing was performed across multiple cohorts, it is likely that further development of EpiScores in larger proteomic samples with diverse ancestries will improve power to generate robust scores. Upper bounds for DNAm prediction of complex traits,

such as proteins, can be estimated by variance components analyses (*Hillary et al., 2020b*; *Trejo Banos et al., 2020*; *Zhang et al., 2019*). Finally, associations present between EpiScore measures and disease incidence may have been influenced by external factors such as prescription medications for comorbid conditions and comorbid disease prevalence.

We have shown that EpiScores for circulating protein levels predict the incidence of multiple diseases, up to 14 years prior to diagnosis. Our findings suggest that DNAm phenotyping approaches and data linkage to electronic health records in large, population-based studies have the potential to (1) capture inter-individual variability in protein levels; (2) predict incident disease risk many years prior to morbidity onset; and (3) highlight disease-relevant biological signals for further exploration. The EpiScore weights are publicly available, enabling any cohort with Illumina DNAm data to generate them and to relate them to various outcomes. Given the increasingly widespread assessment of DNAm in cohort studies (*McCartney et al., 2020*; *Min et al., 2021*), EpiScores offer an affordable and consistent (i.e., array-based) way to utilise these signatures. This information is likely to be important in risk stratification and prevention of age-related morbidities.

# Materials and methods
## The KORA sample population
The KORA F4 study includes 3080 participants who reside in Southern Germany. Individuals were between 32 and 81 years of age when recruited to the study from 2006 and 2008. In the current study, there were 944 individuals with methylation, proteomics, and genetic data available. The Infinium HumanMethylation450 BeadChip platform was used to generate DNAm data for these individuals. The Affymetrix Axiom array was used to generate genotyping data and the SOMAscan platform was used to generate proteomic measurements in the sample.

## DNAm in KORA
Methylation data were generated for 1814 individuals (*Petersen et al., 2014*); 944 also had protein and genotype measurements available. During preprocessing, 65 SNP probes were excluded and background correction was performed in minfi (*Aryee et al., 2014*). Samples with a detection rate of less than 95% were excluded. Next, the minfi R package was used to perform normalisation on the intensity methylation measures (*Aryee et al., 2014*), with a method consistent with the Lumi:QN + BMIQ pipeline. After excluding non-cg sites and CpGs on sex chromosomes or with fewer than 100 measures available, 470,837 CpGs were available for analyses.

## Proteomics in KORA
The SOMAscan platform (Version 3.2) (*Gold et al., 2010*) was used to quantify protein levels in undepleted plasma for 1129 SOMAmer probes (*Suhre et al., 2017*). Of the 1000 samples provided for analysis, two samples were excluded due to errors in bio-bank sampling and one based on quality control (QC) measures. Of the 997 samples available, there were 944 individuals with methylation and genotypic data. Of the 1129 probes available, five failed the QC, leaving a total of 1124 probes for the subsequent analysis. Protein measurements were transformed by rank-based inverse normalisation and regressed onto age, sex, known pQTLs, and 20 genetic principal components of ancestry derived from the Affymetrix Axiom Array to control for population structure. pQTLs for each protein were taken from a previous GWAS in the sample (*Suhre et al., 2017*).

## The LBC1936 and LBC1921 sample populations
The Lothian Birth Cohorts of 1921 (LBC1921; N = 550) and 1936 (LBC1936; N = 1091) are longitudinal studies of aging in individuals who reside in Scotland (*Deary et al., 2012*; *Taylor et al., 2018*). Participants completed an intelligence test at age 11 years and were recruited for these cohorts at mean ages of 79 (LBC1921) and 70 (LBC1936). They have been followed up triennially for a series of cognitive, clinical, physical, and social data, along with blood donations that have been used for genetic, epigenetic, and proteomic measurement. DNAm, proteomic (Olink platform), and genetic data for individuals from Waves 1 (n=875 at mean age 70 years and sd 0.8) and 2 (n=706 at mean age 73 years and sd 0.7) of the LBC1936 and Wave 3 of the LBC1921 (n=162 at mean age 87 years and sd 0.4) were available.

## DNAm in LBC1936 and LBC1921

DNA from whole blood was assessed using the Illumina 450 K methylation array. Details of QC have been described elsewhere (*Shah et al., 2014*; *Zhang et al., 2018*). Raw intensity data were background-corrected and normalised using internal controls. Manual inspection resulted in the removal of low-quality samples that presented issues related to bisulphite conversion, staining signal, inadequate hybridisation, or nucleotide extension. Probes with low detection rate <95% at p< 0.01 and samples with low call rates (<450,000 probes detected at p < 0.01) were removed. Samples were also removed if they had a poor match between genotype and SNP control probes, or incorrect DNAm-predicted sex.

## Proteomics in LBC1936 and LBC1921

Plasma samples were analysed using either the Olink neurology 92-plex or the Olink inflammation 92-plex proximity extension assays (Olink Bioscience, Uppsala Sweden). One inflammatory panel protein (BDNF) failed QC and was removed. A further 21 proteins were removed, as over 40% of samples fell below the lowest limit of detection. Two neurology proteins, MAPT and HAGH, were excluded due to >40% of observations being below the lower limit of detection. This resulted in 90 neurology (LBC1936 Wave 2) and 70 inflammatory (LBC1936 Wave 1) proteins in LBC1936 and 92 neurology proteins available in LBC1921. Protein levels were rank-based inverse normalised and regressed onto age, sex, four genetic components of ancestry derived from multidimensional scaling of the Illumina 610-Quadv1 genotype array and Olink array plate. In LBC1936, pQTLs were adjusted for, through reference to GWAS in the samples (*Hillary et al., 2019*; *Hillary et al., 2020b*).

## Generation Scotland and STRADL sample populations

Generation Scotland: the Scottish Family Health Study (GS) is a large, family-structured, population-based cohort study of >24,000 individuals from Scotland (mean age 48 years) (*Smith et al., 2013*). Recruitment took place between 2006 and 2011 with a clinical visit where detailed health, cognitive, and lifestyle information was collected along with biological samples (blood, urine, saliva). In GS, there were 9537 individuals with DNAm and phenotypic information available. The STRADL cohort is a subset of 1188 individuals from the GS cohort who undertook additional assessments approximately 5 years after the study baseline (*Navrady et al., 2018*).

## DNAm in Generation Scotland and STRADL

In the GS cohort, blood-based DNAm was generated in two sets using the Illumina EPIC array. Set 1 comprised 5190 related individuals whereas Set 2 comprised 4583 individuals, unrelated to each other and to those in Set 1. During QC, probes were removed based on visual outlier inspection, bead count <3 in over 5% of samples, and samples with detection p-value below adequate thresholds (*McCartney et al., 2018b*; *Seeboth et al., 2020*). Samples were removed based on sex mismatches, low detection p-values for CpGs and saliva samples and genetic outliers (*Amador et al., 2015*). The quality-controlled dataset comprised 9537 individuals ($n_{Set1}$ = 5087, $n_{Set2}$ = 4450). The same steps were also applied to process DNAm in STRADL.

## Proteomics in STRADL

Measurements for 4235 proteins in 1065 individuals from the STRADL cohort were recorded using the SOMAscan technology; 793 epitopes matched between the KORA and STRADL cohorts and were included for training in KORA and testing in STRADL. There were 778 individuals with proteomics data and DNAm data in STRADL. Protein measurements were transformed by rank-based inverse normalisation and regressed onto age, sex, and 20 genetic principal components (derived from multidimensional scaling of genotype data from the Illumina 610-Quadv1 array).

## Electronic health data linkage in Generation Scotland

Over 98% of GS participants consented to allow access to electronic health records via data linkage to GP records (Read 2 codes) and hospital records (ICD codes). Data are available prospectively from the time of blood draw, yielding up to 14 years of linkage. We considered incident data for 12 morbidities. Ten of the diseases are listed by the World Health Organization (WHO) as leading causes of either morbidity or mortality (*Hay et al., 2017*; *World Health Organization, 2018*). Inflammatory bowel disease (IBD) (*Kassam et al., 2014*) and RA (*James et al., 2018*) are also included as traits

as they have been reported as leading causes of disability and morbidity and the global burdens of these diseases are rising (*Alatab et al., 2020*; *Safiri et al., 2019*). Prevalent cases (ascertained via retrospective ICD and Read 2 codes or self-report from a baseline questionnaire) were excluded. For IBD prevalent cases were excluded based on data linkage alone. Included and excluded terms can be found in *Supplementary files 1Q-1B1*. Alzheimer's dementia was limited to cases/controls with age of event/censoring ≥65 years. Breast cancer was restricted to females only. Recurrent, major and moderate episodes of depression were included in depression. Diabetes was comprised of predominantly type 2 diabetes codes and additional general diabetes codes such as diabetic retinopathy and diabetes mellitus with renal manifestation that often occur in individuals with type 2 diabetes. Type 1 and juvenile diabetes cases were excluded.

## Elastic net protein EpiScores

Penalised regression models were generated for 160 proteins in LBC1936 and 793 proteins in KORA using Glmnet (Version 4.0-2) (*Friedman et al., 2010*) in R (Version 4.0) (*R Development Core Team, 2020*). Protein levels were the outcome and there were 428,489 CpG features per model in the LBC1936 training and 397,630 in the KORA training. An elastic net penalty was specified (alpha = 0.5) and cross validation was applied. DNAm and protein measurements were scaled to have a mean of zero and variance of one.

In the KORA analyses, 10-fold cross validation was applied and EpiScores were tested in STRADL (n = 778). Of 480 EpiScores that generated ≥1 CpG features, 84 had Pearson r > 0.1 and p < 0.05 in STRADL. As test set comparisons were not available for every protein in the LBC1936 analyses, a holdout sample was defined, with two folds set aside as test data and 10-fold cross validation carried out on the remaining data ($n_{train}$ = 576, $n_{test}$ = 130 for neurology and $n_{train}$ = 725, $n_{test}$ = 150 for inflammatory proteins). We retained 36 EpiScores with Pearson *r* > 0.1 and p < 0.05. New predictors for these 36 proteins were then generated using 12-fold cross validation and tested externally in STRADL (n = 778) and LBC1921 (n = 162, for the neurology panel). Twenty-one EpiScores had *r* > 0.1 and p < 0.05 in at least one of the external test sets. Four EpiScores did not have external comparisons and were included based on holdout performance.

Functional annotations for each of the proteins used to train the finalised set of 109 EpiScores were sourced from the STRING database (*Jensen et al., 2009*). GeneSet enrichment analysis against protein-coding genes was performed using the FUMA database, to quantify which canonical pathways were most commonly implicated across the 109 genes corresponding to the proteins used to train the 109 EpiScores (*Watanabe et al., 2017*). The background gene-set was specified as protein coding genes and a threshold of FDR p < 0.05 was applied for enrichment status, with the minimum overlapping genes with gene-sets set to ≥2.

The 109 selected EpiScores were then applied to Generation Scotland (n = 9537). DNAm at each CpG site was scaled to have a mean of zero and variance of one, with scaling performed separately for GS sets.

## Associations with health linkage phenotypes in Generation Scotland

Mixed effects Cox proportional hazards regression models adjusting for age, sex, and methylation set were used to assess the relationship between 109 EpiScores and 12 morbidities in Generation Scotland. Models were run using coxme (*Therneau, 2020b*) (Version 2.2-16) with a kinship matrix accounting for relatedness in Set 1. Cases included those diagnosed after baseline who had died, in addition to those who received a diagnosis and remained alive. Controls were censored if they were disease free at time of death, or at the end of the follow-up period. EpiScore levels were rank-base inverse normalised. Fully adjusted models included the following additional covariates measured at baseline: alcohol consumption (units consumed in the previous week); deprivation assessed by the Scottish Index of Multiple Deprivation (*GovScot, 2016*); BMI (kg/m$^2$); educational attainment (an 11-category ordinal variable); and a DNAm-based score for smoking status (*Bollepalli et al., 2019*). A false discovery rate multiple testing correction p < 0.05 was applied to the 1308 EpiScore-disease associations (109 EpiScores by 12 incident disease traits). Proportional hazards assumptions were checked through Schoenfeld residuals (global test and a test for the protein-EpiScore variable) using the coxph and cox.zph functions from the survival package (*Therneau, 2020a*) (Version 3.2-7). For each association failing to meet the assumption (Schoenfeld residuals p < 0.05), a sensitivity analysis was run across yearly follow-up intervals.

Fully adjusted Cox proportional hazards models were run with Houseman-estimated white blood cell proportions as covariates (*Houseman et al., 2012*). A further sensitivity analysis added GrimAge

acceleration (*Lu et al., 2019*) as an additional covariate. Basic and fully adjusted Cox models were also run with estimated monocyte, B-cell, CD4T, CD8T, and natural killer cell proportions as predictors, in addition to models with GrimAge acceleration as the predictor of incident disease.

Correlation structures for EpiScores, DNAm-estimated white blood cell proportions, and phenotypic information were assessed using Pearson correlations and pheatmap (*Kolde, 2019*) (Version 1.0.12) and ggcorrplot packages (Version 0.1.3) (*Kassambara, 2019*). The psych package (Version 2.0.9) (*Revelle, 2020*) was used to perform principal components analysis on EpiScores. *Figures 1 and 2* were created with BioRender.com. Associations for EpiScores that were related to a minimum of three morbidities were subset from the fully adjusted Cox proportional hazards results and were visualised using the ggraph package (Version 2.0.5) (*Pedersen, 2021*). This network representation was used (*Figure 5*) to highlight protein EpiScores that were connected with multiple morbidities.

## Consistency of disease associations between EpiScores and measured proteins

Comparisons were conducted between EpiScore-diabetes associations and type 2 diabetes associations with measured proteins using three previous large-scale proteomic studies (*Elhadad et al., 2020*; *Gudmundsdottir et al., 2020*; *Ngo et al., 2021*) In these studies, six cohorts were included (Study 1: KORA n = 993, HUNT n = 940 [*Elhadad et al., 2020*], Study 2: AGES-Reykjavik n = 5438 and QMDiab n = 356 [*Gudmundsdottir et al., 2020*], Study 3: Framingham Heart Study n = 1618 and the Malmo Diet and Cancer Study n = 1221). Study 1 included the KORA dataset, which we use in this study to generate SOMAscan EpiScores. We characterised which SOMAscan-based EpiScore-diabetes associations from our fully adjusted results reflected those observed with measured protein levels. We included basic (nominal $p < 0.05$) and fully adjusted results (with either FDR or Bonferroni-corrected $p < 0.05$), wherever available, across the lookup cohorts (*Supplementary file 1M*).

## Relationship between EpiScores and COVID-19 outcomes

Associations between each of the 109 selected protein EpiScores and subsequent long-COVID or COVID-19 hospitalisation were tested in the Generation Scotland population. A binary variable was used for long-COVID based on self-reported COVID-19 duration from the CovidLife study survey 3 questionnaire (N = 2399 participating individuals) (*Fawns-Ritchie et al., 2021*). Participants were asked about the total overall time they experienced symptoms in their first/only episode of illness, as well as their COVID-19 illness duration. The dataset is correct as of February 2021 when the survey 3 was administered. Of the 9537 individuals with DNAm that were included in incident disease analyses, 173 indicated that they had COVID-19 and 56 of these individuals reported having long-COVID (>4 weeks duration of symptoms after infection). The mean duration from DNAm measurement to long-COVID for these 56 individuals was 11.2 years (sd 1.2). Hospitalisation information, derived from the Scottish Morbidity Records (SMR01), was used to obtain COVID-19 hospital admissions using ICD-10 codes U07.1 (lab-confirmed COVID-19 diagnosis), and U07.2 (clinically diagnosed COVID-19). This data linkage identified 268 of the 9537 individuals that had COVID-19 diagnoses and 29 had been recorded as being hospitalised due to COVID-19. The mean duration from DNAm measurement to hospitalisation for these 29 individuals was 11.9 years (sd 1.4). Logistic regression models with either hospitalisation or long-COVID status as binary outcomes were used, with the 109 scaled protein EpiScores as the independent variables. Sex and age at COVID testing were included as covariates. The latter was defined as the age at positive COVID-19 test or 1 January 2021 if COVID-19 test data were not available.

## Acknowledgements

We are grateful to all study participants of KORA, LBC1936, LBC1921, and GS for their invaluable contributions to this study. This research was funded in whole, or in part, by the Wellcome Trust [104036/Z/14/Z, 220857/Z/20/Z, 108890/Z/15/Z, 203771/Z/16/Z, 216767/Z/19/Z]. For the purpose of open access, the author has applied a CC BY public copyright licence to any Author Accepted Manuscript version arising from this submission.

# Additional information

## Competing interests

Robert F Hillary: has received consultant fees from Illumina. Riccardo E Marioni: has received speaker fees from Illumina and is an advisor to the Epigenetic Clock Development Foundation. The other authors declare that no competing interests exist.

## Funding

| Funder | Grant reference number | Author |
| --- | --- | --- |
| Wellcome Trust | 108890/Z/15/Z | Danni A Gadd<br>Robert F Hillary |
| Wellcome Trust | 203771/Z/16/Z | Anna J Stevenson |
| Alzheimer's Research UK | ARUK-PG2017B−10 | Daniel L McCartney<br>Riccardo E Marioni |
| Qatar Foundation | Biomedical Research Program at Weill Cornell Medicine | Shaza B Zaghlool<br>Karsten Suhre |
| Qatar National Research Fund | NPRP11C-0115-180010 | Shaza B Zaghlool<br>Karsten Suhre |
| Bundesministerium für Bildung und Forschung | Helmholtz Zentrum München | Christian Gieger<br>Annette Peters<br>Melanie Waldenberger<br>Johannes Graumann |
| Munich Center of Health Sciences | LMUinnovativ | Christian Gieger<br>Annette Peters<br>Melanie Waldenberger<br>Johannes Graumann |
| Bavarian State Ministry of Health and Care | DigiMed Bayern | Christian Gieger<br>Annette Peters<br>Melanie Waldenberger<br>Johannes Graumann |
| NIHR Biomedical Research Centre, Oxford | | Liu Shi |
| Dementias Platform UK | MR/L023784/2 | Liu Shi |
| Medical Research Council | MC_UU_00007/10 | Caroline Hayward |
| Wellcome Trust | 104036/Z/14/Z | Ian J Deary<br>David J Porteous<br>Andrew M McIntosh |
| Wellcome Trust | 220857/Z/20/Z | Andrew M McIntosh |
| Wellcome Trust | 216767/Z/19/Z | Chloe Fawns-Ritchie<br>Cliff Nangle<br>Archie Campbell<br>Robin Flaig<br>Ian J Deary<br>David J Porteous<br>Caroline Hayward<br>Andrew M McIntosh<br>Riccardo E Marioni |
| Chief Scientist Office of the Scottish Government Health Directorates | CZD/16/6 | David J Porteous |
| Scottish Funding Council | HR03006 | David J Porteous |
| Australian Research Council | Fellowship FT200100837 | Allan F McRae |

| Funder | Grant reference number | Author |
|---|---|---|
| Australian Research Council | DP160102400 | Peter M Visscher |
| National Health and Medical Research Council | 1113400 | Peter M Visscher |
| Medical Research Council and Biotechnology and Biological Sciences Research Council | MR/K026992/1 | Ian J Deary |
| Biotechnology and Biological Sciences Research Council | | Ian J Deary |
| Royal Society | Wolfson Research Merit Award | Ian J Deary |
| Chief Scientist Office (CSO) of the Scottish Government's Health Directorates | | Ian J Deary |
| Age UK | (Disconnected Mind project) | Sarah E Harris Ian J Deary Simon R Cox |
| Medical Research Council | G0701120 | Ian J Deary Simon R Cox |
| Biotechnology and Biological Sciences Research Council | BB/F019394/1 | Ian J Deary |
| Sir Henry Dale Fellowship jointly funded by the Wellcome Trust and the Royal Society | 221890/Z/20/Z | Simon R Cox |
| National Institutes of Health | RF1AG073593 | Elliot M Tucker-Drob |
| National Institutes of Health | R01AG054628 | Elliot M Tucker-Drob Ian J Deary Simon R Cox |
| Health Data Research UK | substantive site award | Archie Campbell |
| Medical Research Council | MRC Human Genetics Unit core support | Caroline Hayward |
| Medical Research Council | MR/R024065/1 | Ian J Deary Simon R Cox |
| Medical Research Council | MR/M013111/1 | Ian J Deary Simon R Cox |
| Medical Research Council | G1001245 | Ian J Deary Simon R Cox |
| Australian Research Council | FL180100072 | Peter M Visscher |
| National Health and Medical Research Council | 1010374 | Peter M Visscher |
| National Institutes of Health | P30AG066614 | Elliot M Tucker-Drob |
| National Institutes of Health | P2CHD042849 | Elliot M Tucker-Drob |
| Alzheimer's Society | AS-PG-19b-010 | Riccardo E Marioni |

| Funder | Grant reference number | Author |
|---|---|---|
| University of Edinburgh and University of Helsinki joint PhD programme in Human Genomics | | Yipeng Cheng |

The funders had no role in study design, data collection and interpretation, or the decision to submit the work for publication.

### Author contributions

Danni A Gadd, Conceptualization, Formal analysis, Investigation, Methodology, Project administration, Software, Validation, Visualization, Writing – original draft, Writing – review and editing; Robert F Hillary, Conceptualization, Formal analysis, Investigation, Methodology, Software, Visualization; Daniel L McCartney, Shaza B Zaghlool, Conceptualization, Formal analysis, Investigation, Methodology, Software, Validation; Anna J Stevenson, Investigation, Methodology; Yipeng Cheng, Chloe Fawns-Ritchie, Data curation, Formal analysis; Cliff Nangle, Archie Campbell, Robin Flaig, Sarah E Harris, Rosie M Walker, Liu Shi, Elliot M Tucker-Drob, Christian Gieger, Annette Peters, Melanie Waldenberger, Johannes Graumann, Allan F McRae, Ian J Deary, David J Porteous, Caroline Hayward, Peter M Visscher, Simon R Cox, Kathryn L Evans, Andrew M McIntosh, Data curation, Investigation; Karsten Suhre, Conceptualization, Data curation, Investigation, Methodology; Riccardo E Marioni, Conceptualization, Data curation, Formal analysis, Funding acquisition, Investigation, Methodology, Project administration, Resources, Software, Supervision, Writing – original draft, Writing – review and editing

### Author ORCIDs

Danni A Gadd (iD) http://orcid.org/0000-0001-6398-5407
Cliff Nangle (iD) http://orcid.org/0000-0001-5432-1158
Sarah E Harris (iD) http://orcid.org/0000-0002-4941-5106
Karsten Suhre (iD) http://orcid.org/0000-0001-9638-3912
Riccardo E Marioni (iD) http://orcid.org/0000-0003-4430-4260

### Ethics

Human subjects: All KORA participants have given written informed consent and the study was approved by the Ethics Committee of the Bavarian Medical Association. All components of GS received ethical approval from the NHS Tayside Committee on Medical Research Ethics (REC Reference Number: 05/S1401/89). GS has also been granted Research Tissue Bank status by the East of Scotland Research Ethics Service (REC Reference Number: 20/ES/0021), providing generic ethical approval for a wide range of uses within medical research. Ethical approval for the LBC1921 and LBC1936 studies was obtained from the Multi-Centre Research Ethics Committee for Scotland (MREC/01/0/56) and the Lothian Research Ethics committee (LREC/1998/4/183; LREC/2003/2/29). In both studies, all participants provided written informed consent. These studies were performed in accordance with the Helsinki declaration.

### Decision letter and Author response

Decision letter https://doi.org/10.7554/eLife.71802.sa1
Author response https://doi.org/10.7554/eLife.71802.sa2

## Additional files

### Supplementary files

• Supplementary file 1. Demographic information and supplementary datasets. (**A**) Demographic and array information for the cohorts and samples used in the study. (**B**) SomaScan panel EpiScore performance in the Stratifying Resilience and Depression Longitudinally (STRADL) test set. (**C**) Performance of Olink panel EpiScores in holdout, STRADL, and LBC1921 test sets. (**D**) Annotations for the proteins corresponding to the 109 selected EpiScores. (**E**) Predictor weights for the 109 EpiScores from Olink and SomaScan panels which passed testing in independent cohorts. (**F**) CpG feature counts for the 109 selected EpiScores. (**G**) Frequency of CpG sites selected for EpiScores with EWAS catalog annotations to phenotypic traits. (**H**) FUMA canonical pathway Gene set

enrichment for the genes encoding the 109 proteins EpiScores were trained on. (**I**) Basic Cox proportional hazards model results in Generation Scotland. (**J**) Fully adjusted and sensitivity analyses results for Cox proportional hazards models in Generation Scotland. (**K**) Schoenfeld residual Cox sensitivity analyses. (**L**) Schoenfeld residual Cox sensitivity analyses split by year of follow-up. (**M**) SOMAscan-EpiScore diabetes association lookup against three large-scale plasma protein-diabetes studies. (**N**) White blood cell sensitivity analyses. (**O**) GrimAge sensitivity analyses. (**P**) COVID-19 analyses. Q-1B1 Primary and secondary diagnosis codes for each of the 12 morbidities in this study that were used to assign case/control status of participants.

- Transparent reporting form

### Data availability

Datasets generated in this study are made available in Supplementary file 1; this file includes the protein EpiScore weights for the 109 EpiScores we provide for future studies to use. Our Methyl-DetectR shiny app (Hillary and Marioni, 2020) has CpG weights for the 109 EpiScores integrated such that it automates the process of score generation for any DNAm dataset and is available at: https://www.ed.ac.uk/centre-genomic-medicine/research-groups/marioni-group/methyldetectr. A video on how to use the MethylDetectR shiny app to generate EpiScores is available at: https://youtu.be/65Y2Rv-4tPU. All datasets used to create figures are included in Supplementary file 1 and specific locations for these are noted in figure legends. All code used in the analyses is available with open access at the following Gitlab repository: https://github.com/DanniGadd/EpiScores-for-protein-levels (copy archived at swh:1:rev:a5130fab3895a0d95f0dcc8826aa9fb5e8c0fa86). The source data-sets analysed during the current study are not publicly available due to them containing information that could compromise participant consent and confidentiality. Data can be obtained from the data owners. Instructions for Lothian Birth Cohort data access can be found here: https://www.ed.ac.uk/lothian-birth-cohorts/data-access-collaboration. Dr Simon Cox must be contacted to obtain a Lothian Birth Cohort 'Data Request Form' by email: simon.cox@ed.ac.uk. Instructions for accessing Generation Scotland data can be found here: https://www.ed.ac.uk/generation-scotland/for-researchers/access; the 'GS Access Request Form' can be downloaded from this site. Completed request forms must be sent to access@generationscotland.org to be approved by the Generation Scotland access committee. Data from the KORA study can be requested from KORA-gen: https://www.helmholtz-munich.de/en/kora/for-scientists/cooperation-with-kora/index.html. Requests are submitted online and are subject to approval by the KORA board.

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
