## [Editor Report]

This is an important study that demonstrates the potential utility of the circulating proteome for disease prediction and risk stratification.

---

## [Decision Letter]

**Decision letter after peer review:**

Thank you for submitting your article "Epigenetic scores for the circulating proteome as tools for disease prediction" for consideration by *eLife*. Your article has been reviewed by 2 peer reviewers, and the evaluation has been overseen by a Reviewing Editor and Y M Dennis Lo as the Senior Editor. The following individual involved in review of your submission has agreed to reveal their identity: Luigi Ferruccilu (Reviewer #1).

1. The authors should explain why producing methylation score for proteins add information to using proteins directly. Proteins are effectors and physicians are accustomed to using them. The development of GrimAge showed something quite peculiar, namely that the score developed for smoking was predictive of mortality even when the analysis was adjusted for smoking. This suggests that methylation score point to "mechanisms" that are acting on health beyond what is revealed by the variable that was used for initial discovery. Could this hypothesis be tested in this study? It would potentially justify developing methylation score even when information on proteins is available.

2. It is not very clear, at least to one of the reviewers, the value of the network analysis that is reported in Figure 5. Could this be explain both in terms of it technical development and in terms of what such analysis adds to the message of this paper?

3. While the authors touch on the possible clinical values of the analysis reported here, it would be great to read what is the view of the authors in more details and indicate what they suggest should the next step to translate their findings to the clinic.

4. The list of marker proteins is interesting, but it would be useful to carry out their functional clustering for a more meaningful discussion.

*Reviewer #1:*

The manuscript is written in a comprehensible language that should understandable both to the researchers in this filed but also for scientists physicians who can gather beyond the details the potential for this science to change our current approach to medical diagnosis.

The authors used penalized regression to developed methylation scores that predict a large number of circulating proteins in two separate high quality datasets. Than used these scores to predict the onset of major medical condition in a third, independent large database, including information on genetic background and many important confounders. They found that more than 100 of theses scores significantly predicted the development of selected chronic diseases with sufficient effect size to be meaningful, and the prediction was independent of circulating proteins and potential confounders. The level of prediction was not sufficient for medical diagnosis but more than enough to alert a clinician that a patients has a high risk of a developing a specific disease and should be looked at more carefully.

The manuscript is well done, with a rigorous analysis. I have few suggestion that may improve the robustness of the results and also increase the clinical meaningfulness of the results.

1. The authors should explain why producing methylation score for proteins add information to using proteins directly. Proteins are effectors and physicians area accustomed to using them. The development of GrimAge showed something quite peculiar, namely that the score developed for smoking was predictive of mortality even when the analysis was adjusted for smoking. This suggests that methylation score point to "mechanisms" that are acting on health beyond what is revealed by the variable that was used for initial discovery. Is this hypothesis could be tested in this study, it would justify developing methylation score even when information on proteins is available.

2. It is not very clear, at least to me, the value of the network analysis that is reported in Figure 5. Could this be explain both in terms of it technical development and in terms of what such analysis adds to the message of this paper?

3. While the authors touch on the possible clinical values of the analysis reported here, it would be great to read what is the view of the authors in more details and indicate what they suggest should the next step to translate their findings to the clinic.

*Reviewer #2:*

From a methodological point of view, not satisfied by the relatively small size of test samples with a huge number of analyzed markers, which creates the likelihood of conclusions that are valid only within this study (especially without the necessary correction for multiple comparisons in such cases).

The list of marker proteins is interesting, but it would be useful to carry out their functional clustering for a more meaningful discussion.

---

## [Author Response]

The reviewers have discussed their reviews with one another, and the Reviewing Editor has drafted this to help you prepare a revised submission.1. The authors should explain why producing methylation score for proteins add information to using proteins directly. Proteins are effectors and physicians are accustomed to using them. The development of GrimAge showed something quite peculiar, namely that the score developed for smoking was predictive of mortality even when the analysis was adjusted for smoking. This suggests that methylation score point to "mechanisms" that are acting on health beyond what is revealed by the variable that was used for initial discovery. Could this hypothesis be tested in this study? It would potentially justify developing methylation score even when information on proteins is available.

Contextualising the relevance of protein EpiScores

Firstly, we note that the way reviewer 1 spoke about the EpiScores in their review was very helpful to contextualise the potential use for these scores from a clinical standpoint. We have therefore updated our manuscript with this ethos in mind.

Regarding the points made in comment one, we have updated our introduction on pages 4-5 as follows, to clarify the value EpiScores may bring to disease prediction and clinical decision-making:

Page 4: “As proteins are the primary effectors of disease, connecting the epigenome, proteome and time to disease onset may help to resolve predictive biological signatures.”

Page 4: “A leading epigenetic predictor of biological aging, the GrimAge epigenetic clock incorporates methylation scores for seven proteins along with smoking and chronological age, and is associated with numerous incident disease outcomes independently of smoking (Hillary, Stevenson, et al., 2020; Lu et al., 2019). This suggests there is predictive value gained in utilising DNAm scores relevant to protein levels as intermediaries for predictions. Methylation scores also point towards the pathways that may act on health beyond the protein biomarker that they are trained on. A portfolio of methylation scores for proteins across the circulating proteome could therefore aid in the prediction of disease and offer a different, but additive signal beyond methylation or protein data alone.”

Page 5: “DNAm scores for Interleukin-6 and C-Reactive protein have been found to associate with a range of phenotypes independently of measured protein levels, show more stable longitudinal trajectories than repeated protein measurements, and, in some cases, outperform blood-based proteomic associations with brain morphology (Conole et al., 2020; A. J. Stevenson et al., 2021). This is likely due to DNA methylation representing the accumulation of more sustained effects over a longer period of time than protein measurements, which have often been shown to be highly variable in their levels when measured at multiple time points (Koenig et al., 2003; Liu et al., 2015; Moldoveanu et al., 2000; Shah et al., 2014). DNA methylation scores for proteins could therefore be used to alert clinicians to individuals with high-risk biological signatures, many years prior to disease onset.”

Testing protein EpiScores against measured proteins

In response to the specific question noted above (Could this hypothesis be tested in this study?), we would absolutely want to test protein EpiScores against the measured protein equivalents as a priority.

We have previously published work that shows that EpiScores for traits (such as smoking and BMI) and protein levels (IL6 and CRP) have been able to achieve comparable, or enhanced performance as compared to the measured proteins or phenotypes within the same study populations. References:

– IL6 and CRP EpiScores vs measured proteins, in relation to lifestyle phenotypes (Stevenson et al., 2021) https://pubmed.ncbi.nlm.nih.gov/33595649/

– CRP EpiScore vs the measured protein, in relation to a range of brain imaging measures (Conole et al., 2021 – now accepted in Neurology) https://www.medrxiv.org/content/10.1101/2020.10.08.20205245v1

– BMI trait score (Hamilton et al., 2019) https://www.nature.com/articles/s41366-018-0262-3

– Smoking trait score (Corley et al., 2019) https://www.ncbi.nlm.nih.gov/pmc/articles/PMC6779733/

Like these studies, we would ideally like to have direct comparisons between protein EpiScores and measured protein levels for the associations we test in Generation Scotland. However, we are limited by the lack of equivalent protein measurements in Generation Scotland. We are currently generating ~4,000 peptide measurements that map to ~300 proteins in N=20,000 individuals in this cohort and we will therefore be able to validate how protein EpiScores perform in comparison to measured proteins. Many protein biomarkers and acute-phase inflammatory proteins such as CRP and IL6 can be highly variable in single sampling measurements taken across various timepoints. As methylation is likely to represent a more consistent profile of long-term exposure to biological and environmental stressors in the body than single time-point protein levels, we may see that methylation scores outperform the measured proteins in some cases for disease predictions. We have been trialling this experimental design with two protein biomarkers (GDF15 and Nt-pro-BNP) in ongoing (and unpublished) work. In preliminary results (n=20,000), we see that EpiScores for these protein levels are comparable to (or better than) the measured protein in associations with cognitive ability and lifestyle traits.

However, in the present study we recognise the lack of direct comparison is a limitation. We have ensured this is mentioned in our discussion on page 23:

“Second, future studies should assess paired protein and EpiScore contributions to traits, as inference from EpiScores alone, while useful for disease risk stratification, is not sufficient to determine mechanisms. This may also highlight EpiScores that outperform the measured protein equivalent in disease.”

2. It is not very clear, at least to one of the reviewers, the value of the network analysis that is reported in Figure 5. Could this be explain both in terms of it technical development and in terms of what such analysis adds to the message of this paper?

We apologise that the technical development of this figure was not clear. Figure 5 is a visual representation of the fully-adjusted Cox model results between protein EpiScores and incident diseases. It builds on the individual associations presented in Figures 4 and 6 (that are grouped by disease outcomes) by plotting the associations for EpiScores that were predictive of three or greater morbidities. This provides insight into the EpiScores that may harbour a predictive signal that is relevant across multiple disease states. We feel that this visualisation helps to illustrate pathways that may be connected to the development of multimorbidity states as we age. Many of the most highly connected protein EpiScores (e.g. C5, SELL, MMP9) are trained on proteins that are known to associate with a pro-inflammatory state in the body. The remainder also implicate cell adhesion and migration pathways.

We have updated our methods section as follows on page 33:

“Associations for EpiScores that were related to a minimum of three morbidities were subset from the fully-adjusted Cox proportional hazards results and were visualised using the ggraph package (Version 2.0.5) (Pedersen, 2021). This network representation was used (Figure 5) to highlight protein EpiScores that were connected with multiple morbidities.”

We have updated our discussion as follows on page 20:

“Our results indicated that the set of 109 EpiScores are likely to be heavily enriched for inflammatory, complement system, innate immune system pathways, in addition to extracellular matrix, cell remodelling and cell adhesion pathways. This reinforces previous work linking chronic inflammation and the epigenome (Zaghlool et al., 2020). It also suggests that EpiScores could be useful in the prediction of morbidities that are characterised by pro-inflammatory states. An example of this is the EpiScore for Complement Component 5 (C5), which was associated with the onset of five morbidities, the highest number for any EpiScore (Figure 5). The EpiScore for C5 is likely to reflect the biological pathways occurring in individuals with heightened complement cascade activity and could be utilised to alert clinicians to individuals at high risk of multimorbidity. Elevated levels of C5 peptides have been associated with severe inflammatory, autoimmune and neurodegenerative states (Ma et al., 2019; Mantovani et al., 2014; Morgan and Harris, 2015) and a range of C5-targetting therapeutic approaches are in development (Alawieh et al., 2018; Brandolini et al., 2019; Hawksworth et al., 2017; Hernandez et al., 2017; Morgan and Harris, 2015; Ort et al., 2020).”

If after reviewing the additional context provided the reviewers still feel strongly that this Figure should be removed, we would be happy to oblige and either remove it or add it as a supporting Figure attached to Figure 4. However, we believe that there is additional value in illustrating the associations in the context of potential signals of multimorbidity, rather than only showing them grouped by single disease states. These EpiScores for proteins may alert clinicians to those in a population that are likely to represent the highest risk of multimorbidity. These individuals are likely to have a biological profile reflective of poorer general health.

3. While the authors touch on the possible clinical values of the analysis reported here, it would be great to read what is the view of the authors in more details and indicate what they suggest should the next step to translate their findings to the clinic.

We thank the reviewers for this interesting comment. We have provided details in our response below and updated specific parts of the manuscript to improve the narrative on clinical translatability of these scores.

We would need to extend this research in several ways before these EpiScores can become clinically translatable:

First, we hope to validate a wider set of protein EpiScores. This is something we plan to do with samples that we will generate over the course of the next year, which will allow us to train scores in n=20,000 individuals for roughly 300 additional proteins.

Second, once we have a substantial set of EpiScores, we then hope to train single-trait, disease-specific predictor scores using the EpiScores as inputs. This will derive singular scores that inform on the risk of developing individual diseases. These scores may be projected into any population that has DNA methylation measured, and used to stratify those in a population that are high risk for specific diseases (i.e. in the top 5% of a score distribution).

Third, more work is required before we are able to project these types of EpiScores for an individual onto a reference population. Due to processing differences and batch effects, comparing absolute differences in DNAm across populations is complicated. By comparison, relative differences within a single cohort or population are far easier to analyse robustly. If these issues could be overcome, our scores for proteins, diseases and lifestyle traits could be used by a clinician to highlight personalised risk scores for each individual based on their epigenetic profile. This would need to be accompanied by information on the magnitude of that risk in real terms.

We have made the following adjustment to our discussion based on this on page 23:

“First, we demonstrate that EpiScores carry disease-relevant signals that may be clinically meaningful to delineate early disease risk when comparing relative differences within a cohort. However, projecting a new individual onto a reference set is complicated due to absolute differences in methylation quantification resulting from batch and processing effects.”

4. The list of marker proteins is interesting, but it would be useful to carry out their functional clustering for a more meaningful discussion.

We have interpreted this comment as requiring additional functional information about the relationships that may exist between proteins used to train EpiScores, such that we could group EpiScores into subsets by particular functions.

As a caution for this response, we would like to note that the proteins are derived from separate sources (84 SomaScan and 25 Olink) and assays. As the protein selection for the SomaScan and Olink panels also contains inherent biases towards proteins with certain functions (i.e. neurology-associated, inflammatory-associated, or of likely relevance to health systems), interpretation of any protein clustering is not straightforward since we are likely to amplify such biases.

We have assessed the functional clustering of the 109 proteins through the use of the STRING tool, which summarises likely protein-protein interaction based on a range of databases. This provides a visual summary of projected interactions, in addition to a tabular list of functional annotations for each protein (see the revised Supplementary File 1 – Table 1D – STRING functional annotation column). While this is informative (see Author response image 1 – for the 109 gene names corresponding to proteins), we believe that there are several caveats to consider when interpreting the functional clustering of these proteins.

**Author response image 1. sa2fig1:** 

While this visual informs us on the likely interactions between the proteins, we feel that the interpretation of this plot in the context of the protein EpiScores is somewhat unhelpful. The EpiScores reflect the biological state of an individual respective to the levels of specific proteins, but they do not directly represent the proteins themselves and are potentially a different signal. Therefore, the functional clustering for the original proteins only direct us towards pathways of possible relevance that the EpiScore signature may be capturing, rather than being directly comparable to the EpiScore signal. For this reason, we have chosen to include only the functional annotations for the individual proteins used in the original training, rather than the interaction information in Author response image 1, as EpiScores for proteins may not interact in the same manner as the protein equivalents.In Supplementary File 1D, we have added a “Generalised grouping” column, which provides an indication of top-line functional groupings of each of the proteins.

We have also performed a GeneSet enrichment test for the 109 genes corresponding to the proteins using FUMA. This has allowed for mapping of canonical pathways that are most enriched across the set of 109 proteins. We have added these results to the manuscript (Figure 2 —figure supplement 3 for a visual summary, and Supplementary File 1D for a tabular summary).

The two updates we have made in response to this comment have provided additional confirmation that many of the proteins in the set of 109 are heavily associated with immune-regulatory, complement cascade and innate immune system pathways. There are also many that are involved with metabolic pathways and cell migration/conformation signalling pathways. This has helped to make our discussion of the EpiScores more robust and we have updated our manuscript as follows:

Page 9 (results): “GeneSet enrichment analysis of the original proteins used to train the 109 EpiScores highlighted pathways associated with immune response and cell remodelling, adhesion and extracellular matrix function (Supplementary File 1E).”

Page 31 (methods): “Functional annotations for each of the proteins used to train the finalised set of 109 EpiScores were sourced from the STRING database (Jensen et al., 2009). GeneSet enrichment analysis against protein-coding genes was performed using the FUMA database, to quantify which canonical pathways were most commonly implicated across the 109 genes corresponding to the proteins used to train the 109 EpiScores (Watanabe et al., 2017). The background gene-set was specified as protein coding genes and a threshold of FDR P < 0.05 was applied for enrichment status, with the minimum overlapping genes with gene-sets set to greater than or equal to two.”

Page 20 (discussion): “Our results indicated that the set of 109 EpiScores are likely to be heavily enriched for inflammatory, complement system, innate immune system pathways, in addition to extracellular matrix, cell remodelling and adhesion signalling pathways. This reinforces previous work linking chronic inflammation and the epigenome (Zaghlool et al., 2020b) and suggests that these EpiScores may be useful in the prediction of age-morbidities that are characterised by pro-inflammatory states.”